# WAVELET-PACKET POWERED DEEPFAKE IMAGE DETECTION

## ABSTRACT

As neural networks become able to generate realistic artificial images, they have the potential to improve movies, music, video games and make the internet an even more creative and inspiring place. Yet, at the same time, the latest technology potentially enables new digital ways to lie. In response, the need for a diverse and reliable method toolbox arises to identify artificial images and other content. Previous work primarily relies on pixel-space convolutional neural networks or the Fourier transform. To the best of our knowledge, synthesized fake image analysis and detection methods based on a multi-scale wavelet representation, which is localized in both space and frequency, have been absent thus far. This paper proposes to learn a model for the detection of synthetic images based on the wavelet-packet representation of natural and generative adversarial neural network (GAN)-generated images. We evaluate our method on FFHQ (Karras et al., 2019), CelebA, and LSUN source identification problems and find improved or competitive performance. Our forensic classifier has a small network size and can be learned efficiently. Furthermore, a comparison of the wavelet coefficients from these two sources of images allows an interpretation and identifies significant differences.

## 1 INTRODUCTION

While GANs can improve traffic prediction (Aigner & Körner, 2019), extract useful representations from data (Radford et al., 2015), translate textual descriptions into images (Zhang et al., 2017), transfer scene and style information between images (Zhu et al., 2017), and generate original art (Tan et al., 2017), they can also enable abusive actors to quickly and easily generate potentially damaging, highly realistic fake images, colloquially called deepfakes (Parkin; Frank et al., 2020). As the internet becomes a more prominent virtual public space for political discourse and social media outlet (Applebaum, 2020), deepfakes present a looming threat to its integrity that must be met with techniques for differentiating the real and trustworthy from the fake.

Previous algorithmic techniques for separating real from computer-hallucinated images of people have relied on identifying fingerprints in *either* the spatial (Yu et al., 2019; Marra et al., 2019) or frequency (Frank et al., 2020; Dzanic et al., 2020) domain. However, to the best of our knowledge, no techniques have jointly considered the two domains in a multi-scale fashion. We believe our virtual public spaces and social media outlets will benefit from a growing, diverse toolbox of techniques enabling automatic detection of GAN-generated content, where the introduced wavelet-based approach provides a competitive and interpretable addition.

In this paper, we make the following contributions:

- We present a principled wavelet-packet-based analysis of GAN-generated images. Compared to existing work in the frequency domain, we examine the spatio-frequency properties of GAN-generated content for the first time. We find significant differences between real and synthetic images in both the wavelet-packet mean and standard deviation, with increasing frequency and at the edges.

- To the best of our knowledge, we present the first application and implementation of boundary wavelets for image analysis in the deep learning context. Proper implementation

      of boundary-wavelet filters allows us to share identical network architectures across different wavelet lengths.

- As a result of the aforementioned wavelet-based analysis, we build classifiers to identify image sources. We work with fixed seed values for reproducibility and report mean and standard deviations over five runs whenever possible. Our systematic analysis shows slightly improved or competitive performance.

The source code for our wavelet toolbox, the first publicly available toolbox for fast boundary-wavelet transforms in the python world, and our experiments is available at `https://github.com/wavelet-detection-review`.

## 2 RELATED WORK

### 2.1 GENERATIVE ADVERSARIAL NETS

The advent of GANs (Goodfellow et al., 2014) heralded several successful image generation projects. For a recent review on GAN theory, architecture, algorithms, and applications, we refer the reader to Gui et al. (2020). We highlight the contribution of the progressive growth technique, in which a small network is trained on smaller images then gradually increased in complexity/number of weights and image size, on the ability of optimization to converge at high resolutions of $1024 \times 1024$ pixels (Karras et al., 2018). As a supplement, style transfer methods (Gatys et al., 2016) have been integrated into new style-based generators. The resulting style-GANs have increased the statistical variation of the hallucinated faces (Karras et al., 2019; 2020). Finally, regularization (e.g., using spectral methods), has increased the stability of the optimization process (Miyato et al., 2018). The recent advances have allowed large-scale training (Brock et al., 2019), which in turn improves the quality further.

### 2.2 WAVELETS

Originally developed within applied mathematics (Mallat, 1989; Daubechies, 1992), wavelets are a long-established part of the image analysis and processing toolbox (Taubman & Marcellin, 2002; Strang & Nguyen, 1996; Jensen & la Cour-Harbo, 2001). In the world of traditional image coding, wavelet-based codes have been developed to replace techniques based on the discrete cosine transform (DCT) (Taubman & Marcellin, 2002). Early applications of wavelets in machine learning studied the integration of wavelets into neural networks for function learning (Zhang et al., 1995). Within this line of work, wavelets have previously served as input features (Chen et al., 2015), or as early layers of scatter nets (Mallat, 2012; Cotter, 2020). Deeper inside neural networks, wavelets have appeared within pooling layers using static Haar (Wang et al., 2020c; Williams & Li, 2018) or adaptive wavelets (Wolter & Garcke, 2021). Within the subfield of generative networks, ongoing concurrent research (Gal et al., 2021) is exploring the use of wavelets to improve GAN generated image content.

### 2.3 DEEPFAKE DETECTION

Previous work on deepfake image detection broadly falls into two categories. The first group works in the frequency domain. Projects include the study of natural and deep network generated images in the frequency space created by the DCT, as well as detectors based on Fourier features (Zhang et al., 2019; Durall et al., 2019; Dzanic et al., 2020; Frank et al., 2020; Durall et al., 2020; Giudice et al., 2021). In particular, Frank et al. (2020) visualizes the DCT transformed images and identifies artifacts created by different upsampling methods. We learn that the transformed images are efficient classification features, which allow significant parameter reductions in GAN identification classifiers. Instead of relying on the DCT, Dzanic et al. (2020) studied the distribution of Fourier coefficients for real and GAN-generated images. After noting significant deviations of the mean frequencies for GAN generated- and real-images, classifiers are trained. Similarly, Giudice et al. (2021) studies statistics of the DCT coefficients and uses estimations for each GAN-engine for classification. Dzanic et al. (2020) found high-frequency discrepancies for GAN-generated imagery, built detectors, and spoofed the newly trained Fourier-classifiers by manually adapting the coefficients in Fourier-space. Finally, He et al. (2021) combine 2d-FFT and DCT features and observed improved performance.

The second group of classifiers works in the spatial or pixel domain, among others, (Yu et al., 2019; Wang et al., 2020a;b; Zhao et al., 2021a;b) train (convolutional) neural networks directly on the raw images to identify various GAN architectures. Building upon pixel-CNN Wang & Deng (2021) adds neural attention. According to (Yu et al., 2019), the classifier features in the final layers constitute the fingerprint for each GAN and are interesting to visualize. Instead of relying on deep-learning, (Marra et al., 2019) proposed GAN-fingerprints working exclusively with denoising-filter and mean operations, whereas (Guarnera et al., 2020) computes convolutional traces.

To the best of our knowledge, we are the first to combine both schools of thought by proposing to work directly with a spatio-frequency wavelet packet representation in a multi-scale fashion. As we demonstrate in the next section, wavelet packets allow visualization of frequency information while, at the same time, preserving local information to some degree. Note that Tang et al. (2021) uses the first level of a discrete wavelet transform, this means no multi-scale representation as a pre-processing step to compute spectral correlations between the color bands of an image, which are then further processed to obtain features for a classifier.

## 3 METHODS

"Wavelets are localized waves, instead of oscillating forever, they drop to zero." This observation, in the words of Strang & Nguyen (1996), leads to the first motivation for using wavelet transforms. In contrast to the periodic sine and cosine waves in the Fourier-basis, which are localized in frequency, wavelets are localized in both space and frequency. Since images often contain sharp borders, where pixel values change rapidly, slowly decaying Fourier coefficients caused by such short pulses can fill up the spectrum. Reconstruction of the original pulse depends on the cancellation of most coefficients. The locality of wavelets makes it easier to deal with such pulses.

While previous approaches based on the fast Fourier transform (FFT) by Dzanic et al. (2020) and DCT from Frank et al. (2020) and Giudice et al. (2021) provide abundant frequency space information, the global nature of these bases means all spatial relations are missing. The desire to extend our knowledge by exploiting spatial information is our second reason to work with wavelets. Wavelets allow the representation of hierarchical levels of detail in space and frequency.

Before discussing the mechanics of the fast wavelet transform (FWT) and its packet variant, we present a proof of concept experiment in Figure 1. We investigate the coefficients of the level 3 Haar wavelet packet transform. Computations use 5K $1024 \times 1024$ pixel images from Flickr Faces High Quality (FFHQ) (Karras et al., 2019) and 5K $1024 \times 1024$ pixel images generated by StyleGAN. The leftmost columns of Figure 1 show a sample from both FFHQ and a StyleGAN generated one. Mean wavelet packets appear in the second, and their standard deviation in the third column. Finally, the rightmost column plots the absolute mean packets and standard deviation differences. For improved visibility, we re-scale the absolute values of the packet representation, averaged first over the three color bands, using the natural logarithm $\log_e$.

We find that the mean wavelet packets of GAN-generated images are often significantly brighter. The differences become more pronounced as the frequency increases from the top left to the bottom right. The differences in high-frequency packets independently confirm Dzanic et al. (2020), who saw the most significant differences for high-frequency Fourier-coefficients.

The locality of the wavelet filters allows face-like shapes to appear, but the image edges stand out, allowing us to pinpoint the frequency disparities. We now have an intuition regarding the local origin of the differences. We note that Haar wavelet packets do not use padding or orthogonalization and conclude that the differences stem from GAN generation. For the standard deviations, the picture is reversed. Instead of the GAN the FFHQ-packets appear brighter. The StyleGAN packets do not deviate as much as the data in the original dataset. The observation suggests that the GAN did not capture the complete variance of the original data. Our evidence indicates that GAN-generated backgrounds are of particular monotony across all frequency bands.

In the next section, we discuss how the fast wavelet transform (FWT) and wavelet packets in Figure 1 are computed. For multi-channel color images, we transform each color separately, but for simplicity, we will only consider single channels in the following exposition.

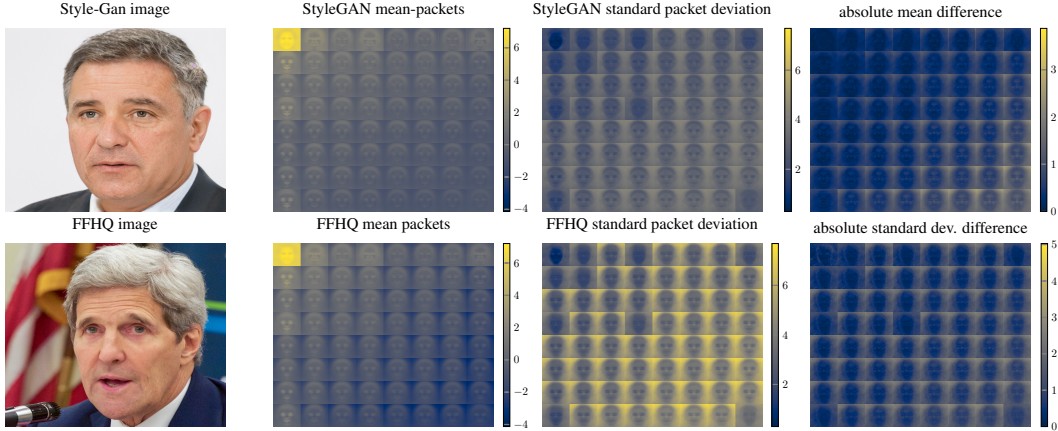

Figure 1: This figure compares single time-domain images and absolute $\log_e$-scaled mean wavelet packets and their standard deviations. The first column shows two example images, the second mean packets, the third standard deviations, and the fourth mean and standard deviation differences. Mean values and standard deviations have been computed on 5K 1024 by 1024 FFHQ- and StyleGAN-generated images each. The order of the packets has the frequency increasing from the top left to the bottom right. We observe significant differences. The mean is significantly different for higher frequencies and at image edges. The standard deviations are different on the background across the entire spectrum. The filter application order of each packet is shown on the right of Figure 2. Best viewed in color.

### 3.1 FAST WAVELET TRANSFORM

Wavelets can be defined by low- and corresponding high-pass filter or by a wavelet function and a scaling function, where we follow the filter construction, as common in signal processing. The FWT (Mallat, 1989; Strang & Nguyen, 1996; Jensen & la Cour-Harbo, 2001) utilizes convolutions to decompose an input signal into its frequency components, where repeated application results in a multi-scale analysis. The resulting wavelet coefficients reflect scale (of level $q$), which is related to frequency, and location, i.e., the position in the transformed and scaled image of level $q$.

Even though 2D wavelets such as the Mexican hat wavelet exist, one dimensional wavelet pairs are often transformed to 2D quadruples using the outer products (Vyas et al., 2018):

$$\mathbf{f}_a = \mathbf{f}_\mathcal{L}\mathbf{f}_\mathcal{L}^T, \qquad \mathbf{f}_h = \mathbf{f}_\mathcal{L}\mathbf{f}_\mathcal{H}^T, \qquad \mathbf{f}_v = \mathbf{f}_\mathcal{H}\mathbf{f}_\mathcal{L}^T, \qquad \mathbf{f}_d = \mathbf{f}_\mathcal{H}\mathbf{f}_\mathcal{H}^T. \qquad (1)$$

Common choices for 1D wavelets are the *Daubechies-wavelets* ("db") and their less asymmetrical variant, the *symlets* ("sym"). We refer the reader to Daubechies (1992) for an in-depth discussion.

In the equations above, $\mathbf{f}$ denotes a filter vector. For one-dimensional filters, the subscripts $\mathcal{L}$ denote the one-dimensional low-pass and $\mathcal{H}$ the high pass filter, respectively. In the two-dimensional case, $a$ denotes the approximation coefficients, $h$ denotes the horizontal coefficients, $v$ denotes vertical coefficients, and $d$ denotes the diagonal coefficients. The 2D transformation at representation level $q+1$ requires the input $\mathbf{x}_q$ as well as a filter quadruple $\mathbf{f}_k$ for $k \in [a, h, v, d]$ and is computed by

$$\mathbf{x}_q * \mathbf{f}_k = k_{q+1}, \qquad (2)$$

where $*$ denotes stride two convolution. Note the input image is at level zero, i.e. $\mathbf{x}_0 = \mathbf{I}$, while for stage $q$ the low pass result of $q-1$ is used as input.

The wavelet transform is invertible. The forward or analysis transform relies on the analysis filters often denoted as $\mathbf{H}_k$ (Strang & Nguyen, 1996). The inverse or synthesis transform filters are typically written as $\mathbf{F}_k$. The down- $\downarrow_2$ and up- $\uparrow_2$ arrow symbols describe the sampling operations in what is known as stride two convolution and transposed convolution in today's machine learning literature. In order to make the transform invertible and plots interpretable, not just any filter will do. The perfect reconstruction and anti-aliasing conditions must hold. We refer to Strang & Nguyen (1996) and Jensen & la Cour-Harbo (2001) for an excellent further discussion of these conditions.

### 3.2 BOUNDARY WAVELETS

So far we described the wavelet transform without considering the finite size of the images. For example, the simple Haar wavelets can be used without modifications in such a case. But, for the transform to preserve all information and be invertible, higher-order wavelets require modifications at the boundary (Strang & Nguyen, 1996). There are different ways to handle the boundary, including zero-padding, symmetrization, periodic extension, and specific filters on the boundary. The disadvantage of zero-padding or periodic extensions is that discontinuities are artificially created at the border, whereas with symmetrization discontinuities of the first derivative arise at the border (Jensen & la Cour-Harbo, 2001). While for large images, the boundary effects might be negligible, for the employed multi-scale approach of wavelet-packets, as introduced in the next subsection, the artifacts become too severe. Furthermore, zero-padding increases the number of coefficients, which in our application would need different neural network architectures per wavelet. Therefore we employ special boundary filters in the form of the so-called Gram-Schmidt boundary filters (Jensen & la Cour-Harbo, 2001).

The idea is now to replace the filters at the boundary with specially constructed, shorter filters that preserve both the length and the perfect reconstruction property or other properties of the wavelet transform. In the appendix Figures 7 and 8 illustrate the sparsity patterns of the resulting sparse analysis and synthesis matrices for the Daubechies two case.

### 3.3 WAVELET PACKETS

Presuming to find the essential information in the lower frequencies, standard wavelet transformations decompose only the low-pass or $a$ coefficients further. The $h$, $v$, and $d$ coefficients are left untouched. While this is often a reasonable assumption (Strang & Nguyen, 1996), previous work (Dzanic et al., 2020) found the higher frequencies equally relevant for deepfake detection. Our analysis, therefore, relies on wavelet packets. For a wavelet packet representation, one recursively continues to filter the low- and high-pass results. Each recursion leads to a new level of filter coefficients, starting with an input image $\mathbf{I} \in \mathbb{R}^{h,w}$, and using $\mathbf{n}_{0,0} = \mathbf{I}$. A node $n_{q,j}$ at position $j$ of level $q$, is convolved with all filters $\mathbf{f}_k$, $k \in [a, h, v, d]$:

$$\mathbf{n}_{q,j} * \mathbf{f}_k = \mathbf{n}_{q+1,k}. \tag{3}$$

Therefore every node at level $q$ will spawn four nodes at the next level $q + 1$. The result at the final level $Q$, assuming Haar or boundary wavelets without padding, will be a $4^Q \times \frac{h}{2^Q} \times \frac{w}{2^Q}$ tensor, i.e. the number of coefficients is the same as before and is denoted by $Q^\circ$. Thereby wavelet packets provide filtering of the input into progressively finer equal-width blocks, with no redundancy. For excellent presentations of the one-dimensional case we again refer to Strang & Nguyen (1996) and Jensen & la Cour-Harbo (2001).

We show a schematic drawing of the process on the left of Figure 2. The upper half shows the analysis transform, which leads to the coefficients we are after. The lower half shows the synthesis transform, which allows inversion for completeness. Finally, for the correct interpretation of Figure 1 the right of Figure 2 lists the filter combinations for packet plots previously shown in Figure 1.

## 4 IMAGE SOURCE SEPARATION EXPERIMENTS

Considering Figure 1 once more, note that the absolute differences of the mean wavelet-coefficients and their standard deviations, shown in the rightmost column, appeared significantly different. In the absolute mean difference plot, we saw widening disparity with increasing frequency along the diagonal. Additionally, background and edge coefficients appeared to diverge. To numerically relate these to each other, we remove the spatial dimensions by averaging over these as well. We observe in the left of Figure 3 increasing mean differences across the board. Differences are especially pronounced at the higher frequencies on the right. In comparison to the FFHQ standard deviation, the variance produced by the StyleGAN, shown in orange, is smaller for all coefficients. In the following sections, we aim to exploit these differences for the identification of artificial images.

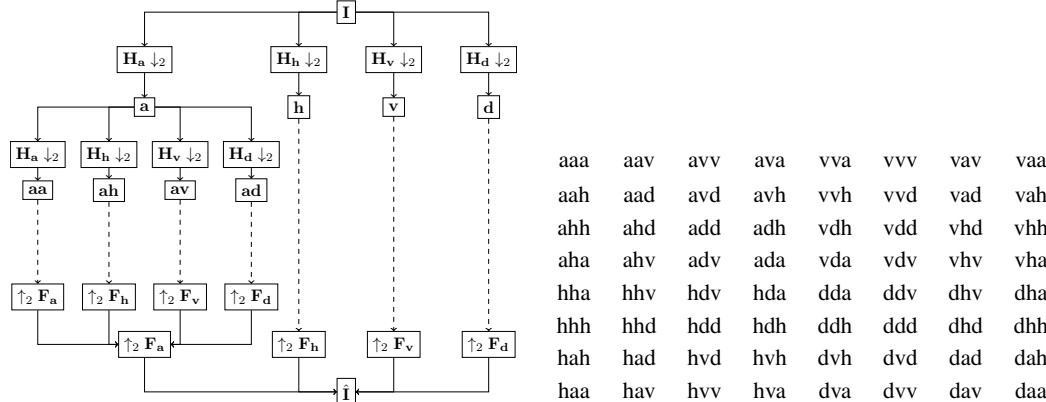

Figure 2: Visualization of the 2D wavelet packet transform analysis and synthesis packet transform (left). The analysis filters are written as $\mathbf{H}_k$, synthesis filters as $\mathbf{F}_k$. We show all first level coefficients as well as some second level coefficients $aa, ah, av, ad$. The dotted lines indicate the omission of further possible analysis and synthesis steps. The transform is invertible in principle, $\hat{\mathbf{I}}$ denotes the reconstructed original input. The right side shows the labels for the transforms of level 3 we showed previously in the frequency order (Jensen & la Cour-Harbo, 2001).

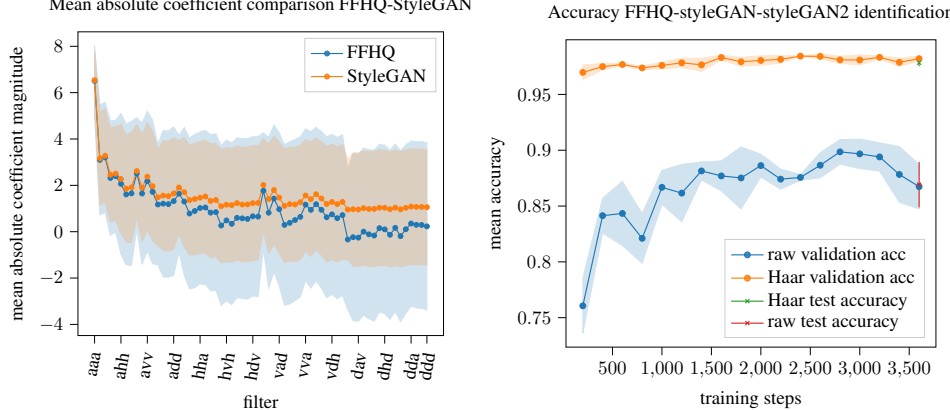

Figure 3: The mean $3°$ Haar wavelet packet coefficient values for each 63K $128 \times 128$ pixel FFHQ (blue) and StyleGAN (orange) images are shown on the left. The shaded area indicates a single standard deviation $\sigma$. We find higher mean coefficient values for the StyleGAN samples across the board. As the frequency increases from left to right, the differences become more pronounced. The plot on the right side shows validation and test accuracy for the identification experiment. Linear regression networks where used to identify FFHQ, StyleGAN, and StyleGAN2 (Karras et al., 2020) images. The blue line shows the pixel and the orange line the training accuracy using $\log_e$-scaled absolute wavelet packet coefficients. Shaded areas indicate a single standard deviation for different initializations. We find that working with $\log_e$-scaled absolute packets allowed linear separation of all three images sources. Furthermore, it significantly improves the convergence speed and final result.

| | Predicted label | | |
|---|---|---|---|
| True label | FFHQ | StyleGAN | StyleGAN2 |
| FFHQ | 4896 | 4 | 100 |
| StyleGAN | 25 | 4973 | 2 |
| StyleGAN2 | 99 | 0 | 4901 |

Table 1: Confusion matrix for the Haar wavelet regression experiment. Overall 98.5% of all samples where classified correctly. Linear Haar-Wavelet regression works particularly well for the StyleGAN. 99.46% of all StyleGAN samples have been properly identified.

## 4.1 FFHQ

Encouraged by the differences we saw in Figures 1 and 3, we attempt to linearly separate the $\log_e$-scaled $3rd$ level Haar wavelet packets by training a linear regression model. We work per class with 63K images for training, 2K for validation, and 5K for testing. All images have a $128 \times 128$ pixel resolution. Wavelet coefficients, as well as raw pixels, are normalized by mean subtraction and division by $\sigma$, using their respective mean and standard deviations computed on the train set. On both normalized features, we train identical networks using the Adam optimizer (Kingma & Ba, 2015) with a step size of 0.001 using PyTorch (Paszke et al., 2017). We plot the mean validation and test accuracy over 5 runs with identical seeds of $0, 1, 2, 3, 4$ in Figure 3 on the right. The shaded areas indicate a single $\sigma$-deviation. On the test set, we observe the confusion table 1. We conclude that in the Haar packet coefficient space, we are able to separate the three sources linearly. Overall, 98.5% of all samples and 99.46% of all StyleGAN images are identified correctly. Observe that working in the Haar-wavelet space improved both the final result and convergence speed.

## 4.2 LARGE-SCALE CELEB FACES ATTRIBUTES AND LARGE-SCALE SCENE UNDERSTANDING

In the previous experiment, essentially two image sources, authentic FFHQ and the related StyleGAN or StyleGAN2 generated images, had to be identified, where we additionally could also distinguish the two artificial sources. In this section, we will study a standardized problem with more image sources. To allow comparison with previous work, we exactly reproduce the experimental setup from Yu et al. (2019) and Frank et al. (2020). Note that we choose this setup since we, in particular want to compare our space and frequency approach with the frequency only DCT-representation. We cite their results for comparison. Additionally, we benchmark against the photoresponse non-uniformity (PRNU) (Marra et al., 2019) approach, as well as the eigenfaces (Sirovich & Kirby, 1987), and straightforward regression as baselines. The experiments in this section use four GANs: CramerGAN (Bellemare et al., 2017), MMDGAN (Binkowski et al., 2018), ProGAN (Karras et al., 2018), and SN-DCGAN (Miyato et al., 2018).

150K images were randomly selected from the Large-scale Celeb Faces Attributes (CelebA) (Liu et al., 2015) and Large-scale Scene UNderstanding (LSUN) bedroom (Yu et al., 2015) datasets. Our pre-processing is identical for both. The real images are cropped and resized to $128 \times 128$ pixels each. With each of the four GANs an additional 150K images are generated at the same resolution. The 750K total images are split into 500K training, 100K validation, and 150K test images. To ensure stratified sampling, we draw the same amount of samples from each generator or the original dataset. As a result, the train, validation, and test sets contain equally many images from each source.

We compute wavelet packets with three levels for each image. We explore the use of Haar and Daubechies wavelets as well as symlets (Daubechies, 1992; Strang & Nguyen, 1996; Jensen & la Cour-Harbo, 2001). The Haar wavelet is also the first Daubechies wavelet. Daubechies-wavelets and symlets are identical up to a degree of 3. We, therefore, start comparing both families above a degree of 4.

For a fair comparison, we normalize both the raw pixels and the wavelet coefficients. Normalization is always the last step before the models. We normalize by subtracting the training-set color-channel mean and dividing each color channel by its standard deviation. The $\log_e$-scaled coefficients are normalized after the rescaling. We train identical CNN-classifiers on top of pixel and various wavelet representations. Additionally, we evaluate eigenface and PRNU baselines. Our wavelet packet transform, regression, and convolutional models are implemented using PyTorch (Paszke et al., 2019). We use Adam (Kingma & Ba, 2015) to optimize our convolutional-models for 20 epochs. The batch size is set to 512 and the learning rate to 0.001. See the supplementary for the exact network layouts.

Table 2 lists the results for comparison. We obtain a beneficial packet representation in every case. On celebA the eigenface approach, in particular, was able to classify *24.11%* more images correctly when run on $\log_e$-scaled Haar-Wavelet packages instead of pixels. For the more complex CNN, Haar wavelets are not enough. More complex wavelets, however, significantly improve the accuracy. With a CNN comparable to Frank et al. (2020), we find accuracy maxima superior to the DCT approach for five wavelets. Demonstrating the robustness of the wavelet approach, the db3 and db4 packets are better on average. On LSUN, we see gains over the raw pixel representation for our $\log_e$-scaled wavelets for CNN as well as eigenface-based classifiers. While we do not outperform the DCT features in this case, we see improvements of the pixel baseline and overall competitive performance.

Table 2: CNN source identification results on the CelebA and LSUN bedroom datasets. We explore the use of boundary-wavelet packets up to a wavelet degree of 5. We report the test set accuracy. Whenever possible we report mean test set accuracy and standard deviation over 5 runs.

| | Accuracy[%] | | | |
| | CelebA | | LSUN bedroom | |
| Method | max | $\mu \pm \sigma$ | max | $\mu \pm \sigma$ |
|---|---|---|---|---|
| Eigenfaces (ours) | 68.56 | - | 62.24 | - |
| Eigenfaces-DCT (Frank et al., 2020) | 88.39 | - | *94.31* | - |
| Eigenfaces-$\log_e$-Haar (ours) | *92.67* | - | 87.91 | - |
| PRNU (ours) | 83.13 | - | 66.10 | - |
| CNN-Pixel (ours) | 98.87 | $98.74 \pm 0.14$ | 97.06 | $95.02 \pm 1.14$ |
| CNN-$\log_e$-Haar (ours) | 97.09 | $96.79 \pm 0.29$ | 97.14 | $96.89 \pm 0.19$ |
| CNN-$\log_e$-db2 (ours) | 99.20 | $98.50 \pm 0.96$ | 98.90 | $98.35 \pm 0.70$ |
| CNN-$\log_e$-db3 (ours) | 99.38 | $99.11 \pm 0.49$ | *99.19* | $99.01 \pm 0.17$ |
| CNN-$\log_e$-db4 (ours) | 99.43 | $99.27 \pm 0.15$ | 99.02 | $98.46 \pm 0.67$ |
| CNN-$\log_e$-db5 (ours) | 99.02 | $98.66 \pm 0.59$ | 98.32 | $97.64 \pm 0.83$ |
| CNN-$\log_e$-sym4 (ours) | 99.43 | $98.98 \pm 0.36$ | 99.09 | $98.57 \pm 0.72$ |
| CNN-$\log_e$-sym5 (ours) | **99.49** | $98.79 \pm 0.48$ | 99.07 | $98.78 \pm 0.24$ |
| CNN-Pixel (Yu et al., 2019) | 99.43 | - | 98.58 | - |
| CNN-Pixel (Frank et al., 2020) | 97.80 | - | 98.95 | - |
| CNN-DCT (Frank et al., 2020) | 99.07 | - | **99.64** | - |

Table 3: Confusion matrix for our best performing CNN using db4-wavelet-packets on CelebA. Classification errors for the original dataset as well as the CramerGAN, MMDGAN, ProGAN, and SN-DCGAN architectures are shown. The test set contains 30,000 entries per label. The detector classifies 99.43% of all images correctly.

| | Predicted label | | | | |
| True label | CelebA | CramerGAN | MMDGAN | ProGAN | SN-DCGAN |
|---|---|---|---|---|---|
| CelebA | 29,759 | 7 | 11 | 107 | 116 |
| CramerGAN | 19 | 29,834 | 144 | 3 | 0 |
| MMDGAN | 17 | 120 | 29,862 | 1 | 0 |
| ProGAN | 136 | 1 | 0 | 29,755 | 108 |
| SN-DCGAN | 55 | 0 | 0 | 5 | 29,940 |

We show the confusion matrix for our best performing network on CelebA in Table 3. Among the GANs we considered the ProGAN-architecture and SN-DCGAN-architecture. Images drawn from both architectures were almost exclusively misclassified as original data and rarely attributed to another GAN. The CramerGAN and MMDGAN generated images were often misattributed to each other, but rarely confused with the original dataset. Of all misclassified real CelebA images most were confused with ProGAN images, making it the most advanced network of the four. Overall we find our, in comparison to a GAN, lightweight 109k classifier able to accurately spot 99.45% of all fakes in this case. Note that all our convolutional networks have only 109,061 parameters, while Yu et al. (2019) used roughly 9 million parameters and Frank et al. (2020) utilized around 170,000 parameters. We observed that using our packet representation improved convergence speed, Figure 10 in the supplementary material illustrates this for the Haar-Wavelet case.

We study the effect of a drastic reduction of the training data size, by reducing the training set size by 80%. Table 4 reports test-accuracies for our retrained classifiers. We find that our Daubechies 4 packet-CNN classifiers are robust to training data reductions, in particular in comparison to the pixel representation.

Table 4: CNN source identification results on the CelebA dataset with only 20 % of the original data. We report the test set accuracy mean and standard deviation over 5 runs as well as the accuracy loss compared to the corresponding network trained on the full data set.

| | Accuracy on CelebA[%] | | | |
| Method | max | Loss | $\mu \pm \sigma$ | Loss |
| --- | --- | --- | --- | --- |
| CNN-Pixel (ours) | 96.37 | -2.50 | $95.16 \pm 0.86$ | -3.58 |
| CNN-$\log_e$-db4 (ours) | **99.01** | -0.41 | $96.96 \pm 3.47$ | -2.58 |
| CNN-Pixel (Frank et al., 2020) | 96.33 | -1.47 | - | - |
| CNN-DCT (Frank et al., 2020) | 98.47 | -0.60 | - | - |

## 5 DETECTION OF IMAGES FROM AN UNKNOWN GENERATOR

To exemplarily study the detection of images from an unknown GAN, we remove the SN-DCGAN generated images from our LSUN training and validation sets. 10,000 SN-DCGAN images now appear exclusively in the test set, where they never influence the learning process. The training hyperparameters are identical to those discussed in section 4.2. We rebalance the data to contain equally many real and gan-generated images. The task now is to produce a real or fake label in the presence of fake images which were not present during the optimization process. For this initial proof-of-concept investigation, we apply only the simple Haar wavelet. We find that our approach does allow detection of the SN-DCGAN samples on LSUN-bedrooms, without their presence in the training set. Using the $\log_e$-scaled Haar-Wavelet packages, we achieve for the unknown GAN a result of $78.8 \pm 1.8\%$, with a max of $81.7\%$. For the real and artificial generators present in the training set, we achieve $98.6 \pm 0.1\%$, with a max of $98.6\%$. The accuracy improvement in comparison to Table 2 likely is due to the binary classification problem here instead of the multi-classification one before.

This result indicates that the wavelet representation alone can allow a transfer to unseen generators, or other manipulations, to some extent. More detailed studies, also with wavelets of higher degree, and in comparison, or together, with transfer learning approaches (Jeon et al., 2020) or autoencoders particularly designed for transferability (Cozzolino et al., 2018) are warranted future work.

## 6 CONCLUSION AND OUTLOOK

This paper presented a wavelet packet-based approach for deepfake analysis and detection, which is based on a multi-scale image representation in space and frequency.

We saw that wavelet-packets allow the visualization of frequency domain differences while preserving some spatial information. We found diverging mean values for packets at high frequencies. At the same time, the bulk of the standard deviation differences were at the edges and within the background portions of the images. This observation suggests contemporary GAN architectures still fail to capture the backgrounds and high-frequency information in full detail. The proposed wavelet packet representations allowed linear separation of the FFHQ-StyleGAN-StyleGAN2 source identification problem. We studied additional wavelets on the CelebA and LSUN-bedroom problems. We found that coupling higher-order wavelets and CNN led to an improved or competitive performance in comparison to a DCT approach or working directly on the raw images. The employed lean neural network architecture allows efficient training and can achieve similar accuracy with only 20% of the training data. Overall our classifiers deliver state-of-the-art performance, require few learnable parameters, and converge quickly.

Even though releasing our detection code will allow potential bad actors to test against it, we hope our approach will complement the publicly available deepfake identification toolkits. We propose to investigate the resilience of multi-classifier ensembles in future work and envision a framework of multiple forensic classifiers, which together give strong and robust results for artificial image identification.

Future work could also explore the use of curvelets and shearlets for deepfake detection.

## 7 REPRODUCIBILITY STATEMENT

To ensure the reproducibility of this research project, we release our wavelet toolbox and the source code for our experiments in the supplementary material. The wavelet toolbox ships multiple unit tests to ensure correctness. Our code is carefully documented to make it as accessible as possible. We have consistently seeded the Mersenne twister used by PyTorch to initialize our neural networks with 0, 1, 2, 3, 4. Since the seed is always set, rerunning our code always produces the same results. Outliers can make it hard to reproduce results if the seed is unknown. To ensure we do not share outliers without context, we report mean and standard deviations of five runs for all neural network experiments.

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

## ACRONYMS

**CelebA** Large-scale Celeb Faces Attributes

**CNN** convolutional neural network

**DCT** discrete cosine transform

**FFHQ** Flickr Faces High Quality

**FFT** fast Fourier transform

**FWT** fast wavelet transform

**GAN** generative adversarial neural network

**LSUN** Large-scale Scene UNderstanding

**PRNU** photoresponse non-uniformity

# 8 SUPPLEMENTARY

In this supplementary, we show the full architectures for all networks. In terms of training convergence, we give the validation accuracy mean and standard deviation during training. Furthermore, we show additional visualizations of the wavelet packets.

## 8.1 NETWORK ARCHITECTURES

| Pixel-CNN | |
|---|---|
| Convolution | (3x8x3x3), |
| ReLU, | |
| Convolution | (8x8x3), |
| ReLU, | |
| Pool | (2, 2), |
| Convolution | (8x16x3x3), |
| ReLU, | |
| Pool | (2, 2), |
| Convolution | (16x32x3x3), |
| ReLU, | |
| Linear | (32*28*28, classes) |

| Wavelet-Packet-CNN | |
|---|---|
| Convolution | (192x24x3x3), |
| ReLU, | |
| Convolution | (24x24x6x6), |
| ReLU, | |
| Convolution | (24x24x9), |
| ReLU, | |
| Linear | (24, classes) |

Table 5: The CNN architectures we used in our experiments. We show the convolution and pooling kernel, as well as matrix sizes in brackets. As the packet representation generates additional channels we are forced to adapt the CNN architecture. The parameter counts vary depending on the number of classes, but has have approximately 100k for all experiments.

Unless explicitly stated otherwise, we train with a batch size of 512 and a learning rate of 0.001 using Adam (Kingma & Ba, 2015) for 20 epochs. For the 5 repetitions we report the seed values are set to 0,1,2,3,4.

## 8.2 PERTURBATION ANALYSIS

We study the effect of image cropping, rotation, and jpeg compression on our classifiers in table 6. We re-trained both classifiers on the perturbed data as described in section 4.2. The center crop perturbation randomly extracts images centers while preserving at least 80% of the original width or height. Both the pixel and packet approaches are affected by center cropping yet can classify more

Table 6: Perturbation analysis of the db3 wavelet-packet- and pixel- CNN classifiers on LSUN-bedrooms.

| Method | Perturbation | Accuracy on LSUN[%] | |
|---|---|---|---|
| | | max | $\mu \pm \sigma$ |
| CNN-$\log_e$-db3 (ours) | center-crop | **95.68** | 95.49± 0.26 |
| CNN-Pixel (ours) | center-crop | 92.03 | 90.04± 1.18 |
| CNN-$\log_e$-db3 (ours) | rotation | 91.74 | 90.84± 0.90 |
| CNN-Pixel (ours) | rotation | **92.63** | 91.99± 0.97 |
| CNN-$\log_e$-db3 (ours) | jpeg | 84.73 | 84.25 ± 00.33 |
| CNN-Pixel (ours) | jpeg | **89.95** | 89.25 ± 00.48 |

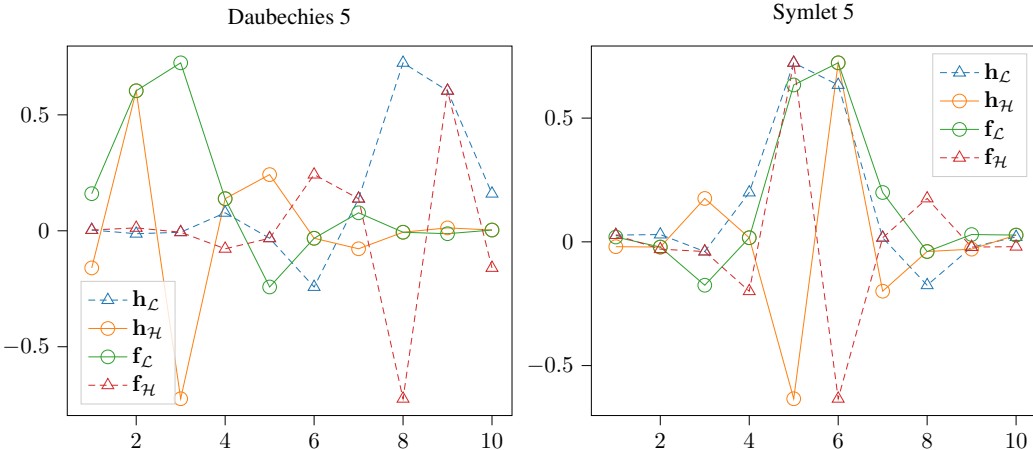

Figure 4: Fifth degree Daubechies and Symlet filters side by side.

than 90% of all images correctly. In this case, the log-scaled db3-wavelet packets can outperform the pixel-space representation on average. For rotation and jpeg compression the picture changes. The rotation perturbation randomly rotates all images by up to $15°$ degrees. The jpeg perturbation randomly compresses images with a compression factor drawn from $U(70, 90)$, without subsampling. The wavelet-packet representation is less robust to rotation than its pixel counterpart. For jpeg-compression, the effect is more pronounced. Looking at Figure 3 suggests that our classifiers rely on high-frequency information. Since jpeg-compression removes this part of the spectrum, perhaps explains the larger drop in mean accuracy.

### 8.3 THE PERFECT RECONSTRUCTION AND ALIAS CANCELLATION CONDITIONS

When using wavelet filters, we expect a lossless representation free of aliasing. Consequently, we want to restrict ourselves to filters, which guarantee both. Classic literature like Strang & Nguyen (1996) presents two equations to enforce this, the perfect reconstruction and alias cancellation conditions. Starting from the analysis filter coefficients $\mathbf{h}$ and the synthesis filter coefficients $\mathbf{f}$. For a complex number $z \in \mathbb{C}$ their z-transformed counterparts are $H(z) = \sum_n \mathbf{h}(n)z^{-n}$ and $F(z)$ respectively. The reconstruction condition can now be expressed as

$$H_{\mathcal{L}}(z)F_{\mathcal{L}}(z) + H_{\mathcal{H}}(z)F_{\mathcal{H}}(z) = 2z^c, \tag{4}$$

and the anti-aliasing condition as

$$H_{\mathcal{L}}(-z)F_{\mathcal{L}}(z) + H_{\mathcal{H}}(-z)F_{\mathcal{H}}(z) = 0. \tag{5}$$

For the perfect reconstruction condition in Eq. 4, the center term $z^l$ of the resulting z-transformed expression must be a two; all other coefficients should be zero. $c$ denotes the power of the center. The effect of the alias cancellation condition can be studied by looking at Daubechies wavelets and symlets, which we will do in the next section. The perfect reconstruction condition is visible too, but harder to see. For an in-depth discussion of both conditions, we refer the interested reader to the excellent textbooks by Strang & Nguyen (1996) and Jensen & la Cour-Harbo (2001).

### 8.4 DAUBECHIES WAVELETS AND SYMLETS

Figure 4 plots both Daubechies filter pairs on the left. A possible way to solve the anti-aliasing condition formulated in equation 5 is to require, $F_{\mathcal{L}}(z) = H_{\mathcal{H}}(-z)$ and $F_{\mathcal{H}}(z) = -H_{\mathcal{L}}(-z)$ (Strang & Nguyen, 1996). The patterns this solution produces are visible in for both the Daubechies filters as well as their symmetric counterparts shown above. To see why substitute $(-z)$. It will produce a minus sign at odd powers in the coefficient polynomial. Multiplication with $(-1)$ shifts the pattern to even powers. Whenever $F_{\mathcal{L}}$ and $H_{\mathcal{H}}$ share the same sign $F_{\mathcal{H}}$ and $H_{\mathcal{L}}$ do not and the other way around. The same pattern is visible for the Symlet on the right of figure 4.

The Daubechies wavelets are very anti-symmetric (Mallat, 2009). Symlets have been designed as an alternative with similar properties. But, as shown in 4, Symlets are symmetric and centered.

analysis Haar level 1          synthesis Haar level 1.tex

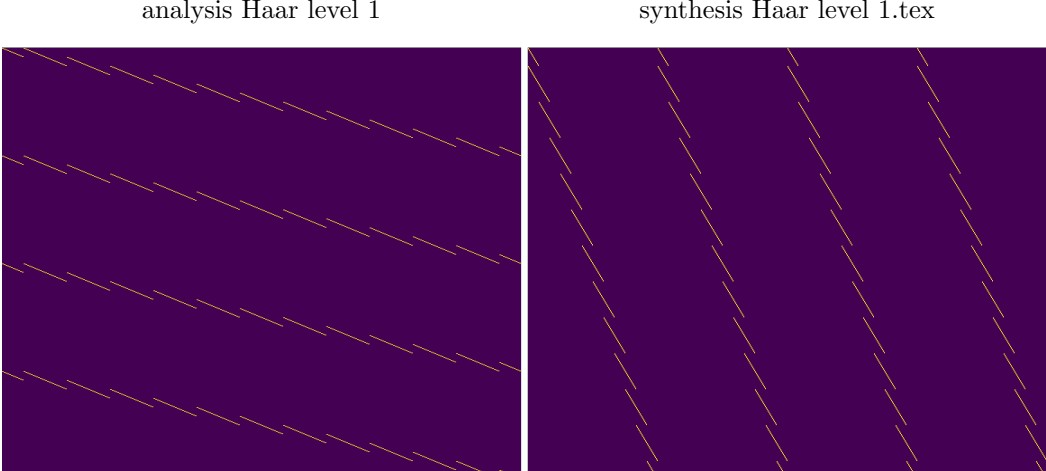

Figure 5: Sparsity pattern of the 2d level 1 Haar fast wavelet analysis and synthesis matrices.

analysis Haar level 2          synthesis haar level 2.tex

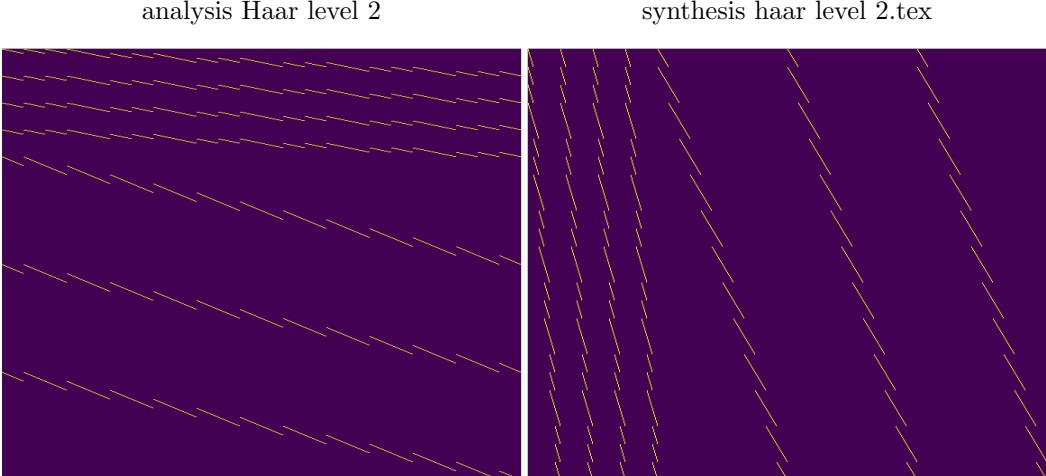

Figure 6: Sparsity pattern of the 2d level 2 Haar fast wavelet analysis and synthesis matrices.

## 8.5 SPARSE FAST WAVELET TRANSFORMATION MATRIX PLOTS

analysis db2 level 1                 synthesis db2 level 1

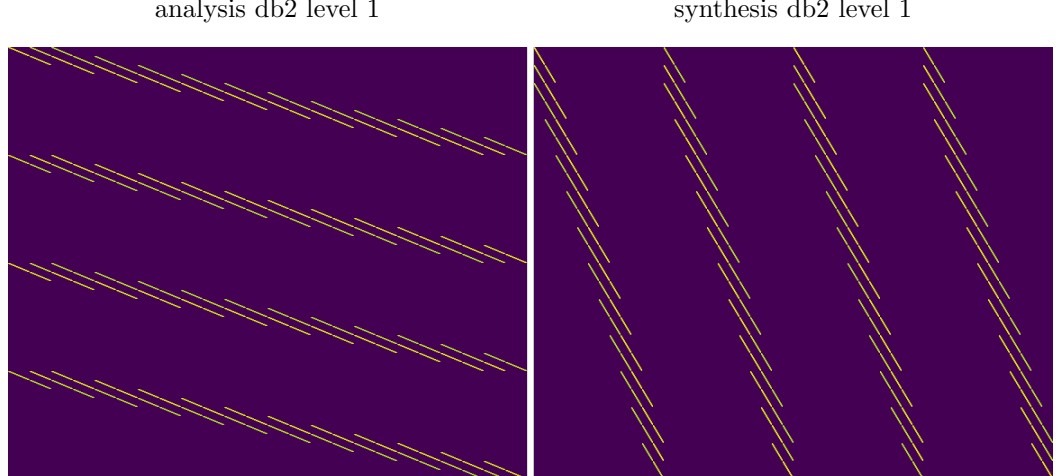

Figure 7: Sparsity pattern of the 2d level 1 Daubechies-2 fast wavelet analysis and synthesis matrices.

analysis db2 level 2                 synthesis db2 level 2

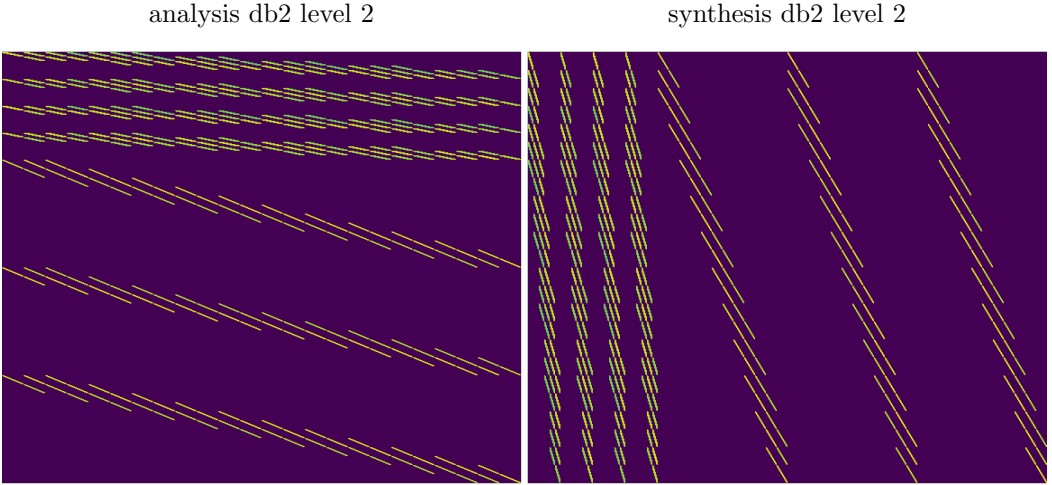

Figure 8: Sparsity pattern of the 2d level 2 Daubechies-2 fast wavelet analysis and synthesis matrices.

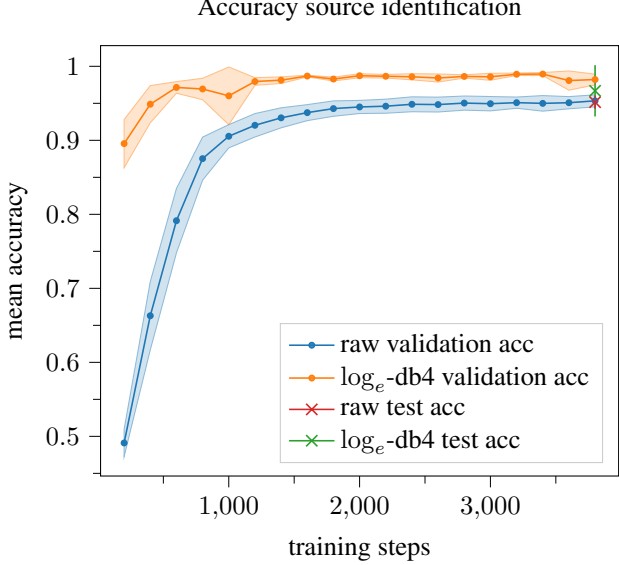

Figure 9: Mean validation and test set accuracy of 5 runs of source identification on CelebA for a CNN trained on the raw images and $\log_e$-db4 wavelet packets, each using only 20 % of the original training data. The shaded areas indicate a single standard deviation $\sigma$.

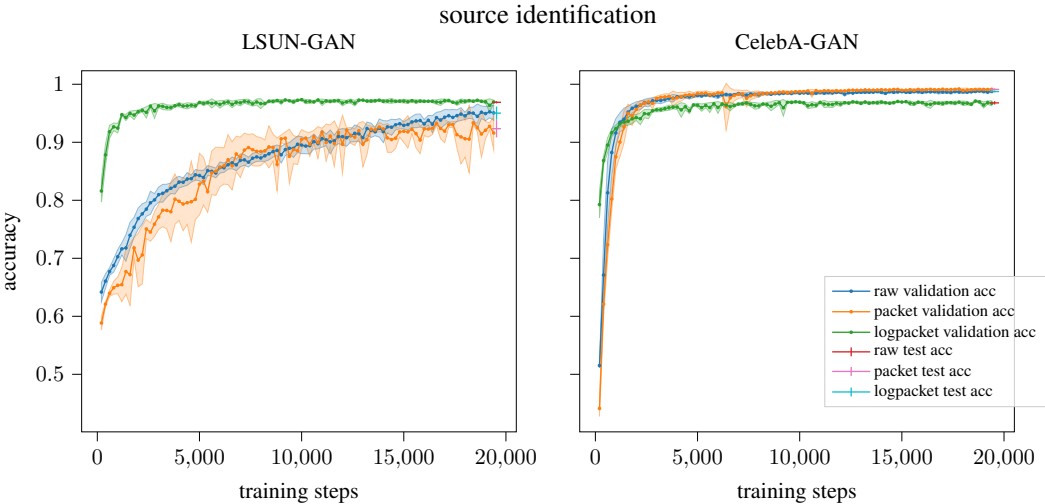

Figure 10: Haar-Wavelet packet CNN training accuracy on the LSUN (left) and the CelebA (right) datasets.

