# OpenReview forum: "Wavelet-Packet Powered Deepfake Image Detection"
_ICLR.cc/2022/Conference — ICLR 2022 Submitted_

### Official Review · Reviewer_SoZC · 2021-11-02

**Correctness:** 3
**Technical Novelty And Significance:** 2
**Empirical Novelty And Significance:** 2
**Recommendation:** 5
**Confidence:** 3

**Details Of Ethics Concerns:**

Not applicable for this one.

**Main Review:**

#Strengths:
1) The study pursued should be pursued with importance given the rise in deepfake images in the internet.
2) The methods section is well written.

#Weaknesses:
The paper can be improved based on the following points:
1) In section 1, it would be better to include a motivational case study to emphasize the importance of the research being pursued for a broader group of readers to be motivated.
2) Please briefly mention why previous works only relied on identifying fingerprints in either the spatial or frequency domain? What were the main challenges associated and why/how is this research work is able to overcome that? This wasn’t mentioned properly in section 1 or 2.
3) In section 3.3, for Fast Wavelet Transform (FWT) please briefly mention the use of FWT and its associated justification. Please do the same for 3.4 i.e. Boundary Wavelet as well.
4) Though in section 3 the different wavelet methods are mentioned well, for the equations the symbols used are not intuitive and explanatory. Please specify them properly.
5) In the experimental section, the comparative study seemed to be weak. Why were the methods chosen? Could you please provide a variety of comparative study to prove efficacy?

**Summary Of The Paper:**

In this paper, the authors propose to learn a model for the detection of synthetic images based on the wavelet-packet representation of natural and generative adversarial neural network (GAN)-generated images. In experimental evaluation, the proposed method was compared with FFHQ, CelebA, and LSUN source identification problems.

**Summary Of The Review:**

The paper is well written in some places, however, does has the opportunity to be improved, especially it was lacking justification in some places and could be helpful to include some more comparative study to prove the efficacy. Please refer to the points mentioned under weaknesses section in the main review to get details.

---

> ### Author Response · Authors · 2021-11-19
> **We thank reviewer SoZC for for an informative and honest review.**
>
> - Quality of the writing:
>
>     We are happy to read SoZC and qN3N found the methods section well written.
>
> 1. Motivational case study:
>
>     We are hoping Figure 1 would serve as a motivation. We have arranged the wavelet packets such, that
>     the frequency increases along the diagonal. A closer look reveals differences between images from stylegan
>     and FFHQ for higher frequencies and for background pixels. Previous Fourier-based methods do not allow
>     spatial interpretation.
>
> 2. Problems with spatial or frequency domain work:
>
>     In principle, Fourier-basis functions oscillate forever. Working with a pure Fourier basis leads to an ideal frequency domain resolution,
>     but all spatial (or temporal) information is lost.
>     Wavelets, in contrast, do not oscillate forever, they drop to zero eventually. This important difference allows wavelet transformations to
>     conserve spatial information to some degree.
>     Purely spatial approaches do not have access to the frequency domain, while Fourier methods lose all spatial information.
>     The motivation of our work is to popularize the wavelet approach to a wider audience within the deep learning community.
>     To achieve this we aim to demonstrate the utility of the boundary-wavelet packet approach for deepfake detection and provide the necessary
>     tools in a documented python package.
>
>
> 3. Motivation for the use of boundary wavelets:
>
>     For the wavelet transform to be correct, we have to apply all wavelet filter coefficients to every pixel. Traditionally image-padding ensures this.
>     However, padding also enlarges the representation we compute. Enlargement leads to additional weights for longer wavelet filters. Boundary filters solve this problem. We construct, truncate and orthogonalize sparse operator matrices to eliminate the padding. As a result, we are able to share precisely the same network architecture for all packet representations we compare.
>
> 4. Intuitive explanation of symbols:
>
>     We have simplified how we state the naming convention for each dimension.
>     We use $\mathcal{L},\mathcal{H}$ for 1d low and highpass filters.
>     In the two-dimensional case we use; a (approximation), h (horizontal detail), v (vertical detail) and d (diagonal detail). We hope its more intuitive now.
>
> 5. Choice of the methods in the experimental section:
>
>     We reproduced the setup es described by Frank et al. (2020)., The idea of table two is to show that older methods like eigenface-classification as well as modern CNN can benefit from wavelet features.

---

> > ### Comment · Reviewer_SoZC · 2021-11-25
> > **Reply to authors' rebuttal**
> >
> > Thank you for explaining some of the points mentioned in the review, however, after going through all the comments my final decision is "marginally below acceptance threshold". I wish you all the best with you paper.

---

### Official Review · Reviewer_zoMp · 2021-11-06

**Correctness:** 4
**Technical Novelty And Significance:** 3
**Empirical Novelty And Significance:** 2
**Recommendation:** 5
**Confidence:** 5

**Main Review:**

Strengths:

Usage of wavelets to detect GAN generated images is a definite strength of this paper. Both pixel domain and frequency domain methods have been applied in the past to detect GAN generated images. Usage of wavelets integrates these.

The source separation experiments are also strengths in this paper.

Weakness:

One main drawback of the paper is that no robustness analysis is done on the proposed method. While the authors mention that they will do this as future work, this analysis is an important part in media forensics and it is essential to know how a detection method would perform when post processing operations are performed on the image. It would have been good if the authors presented the results on at least some post processing operations like JPEG compression and noise addition.

It will also be good to consider more GAN datasets such as CycleGAN, StarGAN, BIGGAN and others, such that the GANs are all diverse.

Many past papers have shown patterns in the frequency domain for GAN generated images that are flagged due to upsampling. It will be good to compare those patterns and the wavelet patterns side by side. Patterns can be found in:

Ref (a). Zhang, Xu, Svebor Karaman, and Shih-Fu Chang. "Detecting and simulating artifacts in gan fake images." In 2019 IEEE International Workshop on Information Forensics and Security (WIFS), pp. 1-6. IEEE, 2019.

Ref (b). Wang, Sheng-Yu, Oliver Wang, Richard Zhang, Andrew Owens, and Alexei A. Efros. "CNN-generated images are surprisingly easy to spot... for now." In Proceedings of the IEEE/CVF Conference on Computer Vision and Pattern Recognition, pp. 8695-8704. 2020.



**Summary Of The Paper:**

This paper presents an approach to detect GAN generated Deepfake images using wavelet features. The paper also showed linear separation of GANs using the proposed approach. Experiments presented on many datasets showed that the approach is promising and presents a method that integrates both spatial and frequency domain information.

**Summary Of The Review:**

While the paper provides an interesting method to detect GAN generated images using wavelets, thus combining pixel and frequency domain, more experiments are needed to provide a good assessment of the method.

---

> ### Author Response · Authors · 2021-11-19
> **We thank reviewer zoMp for the valuable feedback.**
>
> - Potential of the method:
>
>     We are happy to hear that zoMp like qN3N found our method promising.
>
> - Experimental evaluation:
>
>     We have included robustness experiments in table 6. We find some resistance of our approach to center-crop and rotation perturbations.
>
> - Novelty and value to the community:
>
>     Numerous works have shown the utility of Fourier coefficients for deepfake detection (Zhang et al., 2019; Durall et al., 2019; Dzanic et al., 2020; Frank et al., 2020; Durall et al., 2020; Giudice et al., 2021). This paper lays the groundwork for the use of wavelet coefficients for this task.
>     Using boundary wavelet filters requires careful treatment of the sparse fast wavelet operator matrices, which was previously unavailable in the python world. We encourage zoMP to take a look at our source code, the tests as well as the documentation.
>     In addition to the methodology, we present experiments using a total of 7 different GANs on three publicly available datasets with comparisons to two other papers, both of which appeared at respected venues.
>     We have added the requested robustness experiments and argue that enough data showing the utility of the presented approach is now available.
>
> - Up-sampling plots:
>
>     Thank you for the suggestion. We agree. Plots illustrating the effect of upsampling operations in the wavelet-packet representation would be nice to have.
>     Given the time constraints of the rebuttal, we are unable to provide this analysis at this time.

---

### Official Review · Reviewer_feXB · 2021-11-08

**Correctness:** 3
**Technical Novelty And Significance:** 2
**Empirical Novelty And Significance:** 2
**Recommendation:** 3
**Confidence:** 4

**Main Review:**

Strengths:
1. The paper uses a different representation of images i.e wavelet-packets to solve source identification problem which has not been used before.
2. The analysis on using different levels  for Daubechies and symlets wavelets provides better understanding of which level is useful for detecting GAN generated images.

Weaknesses:

1. The main idea of using wavelet coefficients to perform deepfake detection is not a novel approach. Wavelet representation is localized in space and frequency domain. However, both domains have already been studied in the past [1], [2]. Therefore, using a representation combining both of them is not a very exciting idea. Looking at the source identification results in Tab. 2 highlights the fact that the combined representation is performing very similar to previous approaches with worse performance for LSUN dataset. This makes me question the contribution of the proposed approach for the community. Using a new representation although gives an idea to look at a new direction, but the performance doesn't contribute positively for the same.

2. The authors provide a thorough analysis of their approach in Tab. 2 which talks about the source identification task. However, the problem of source identification is quite simple and previous approaches already have made significant advancement in the performance. The authors try to go one step further by removing one GAN from the training set and evaluate their approach. However, they don't provide any baseline results to compare with for proposed testing protocol. This gives no conclusion whether the proposed approach works better than pervious works or not.


3. The paper proposes to use wavelet coefficients at a certain level to perform a transformation of input image using quadruple filters as shown in Eq. 2. The filters defined in Eq. 1 are used to perform the above transformation. These filters denote the low-low, low-high, high-low and high-high coefficients. However, the intuition for using these four types of coefficients is still not clear. Each type of coefficient can have different performance for deepfake detection. It is not clear which of the four is responsible for the reported result in Tab. 2 and 4.

4.  In page 4, there is a discussion for the wavelet transform being invertible. This is done using analysis filters H and inverse filters F. The paragraph further talks about perfect reconstruction and anti-aliasing conditions to select these filters. However, a brief introduction of the above mentioned conditions and how these conditions are used to select filters for the proposed framework should be mentioned clearly.

5. The authors tackle the problem of source identification. But it would be interesting to see how the proposed method would perform when trained and tested on deepfake datasets like FaceForensics++, DFDC, UADFV, etc. Real vs fake binary classification performance would provide much more clarity on whether the proposed approach actually works for deepfake detection or not as depicted by the main title of the paper.

6. In section 4.1, the authors use a linear regression model to separate the 3rd level Haar wavelet packets for classification. However, in section 4.2, the authors use a CNN-classifier to perform classification. it is not clear why different models were used for different datasets.

7. The authors use Haar and Daubechies wavelets as well as symlets wavelets in their experiments. However, it would be beneficial to include a brief introduction for both type of wavelets in section 3 to better understand the intuition behind selecting these two for this work.


8. The setting of experiment on FFHQ images dataset mentions that the images are used at a resolution of 128 x 128. However, FFHQ has 1024 x 1024 images. Same is the output for StyleGAN and StyleGAN2. So my concern is that the patterns observed in the wavelet visualization in Figure 1 might be formed due to downsampling of these images. Therefore, the difference in the visualization might come from downsampling rather than GAN itself.

References:
1. Ning Yu, Larry S Davis, and Mario Fritz. Attributing fake images to gans: Learning and analyzing gan fingerprints. In Proceedings of the IEEE/CVF International Conference on Computer Vision, pp. 7556–7566, 2019
2. Joel Frank, Thorsten Eisenhofer, Lea Schönherr, Asja Fischer, Dorothea Kolossa, and Thorsten Holz. Leveraging frequency analysis for deep fake image recognition. In Proceedings of the 37th International Conference on Machine Learning, ICML 2020, 13-18 July 2020, Virtual Event, volume 119 of Proceedings of Machine Learning Research, pp. 3247–3258. PMLR, 2020.

**Summary Of The Paper:**

This paper proposes a framework to perform source identification using wavelet-packet-based analysis of GAN-generated images. They implement boundary-wavelet filters which make use of the share identical network architectures across different wavelet lengths to perform GAN source classification.

**Summary Of The Review:**

The paper lacks enough novelty in the technical approach. The authors tackle a very simple problem of source identification. The authors try to perform evaluation on unseen GANs but lack of baselines to compare it with makes the analysis redundant. The experimental evaluation is not thorough across different datasets for deepfake detection which seems to be the main motivation of the paper.

---

> ### Author Response · Authors · 2021-11-19
> **We thank Reviewer feXB for taking the time to review our paper.**
>
>
> - Novelty and quality of our study:
>
>     We were happy to see that feXB, like qN3N, and SHXs saw that our paper is the first application of wavelet packets for deepfake analysis and detection. Furthermore, we are delighted to see feXB liked our efforts to study different wavelets.
>
> - Relation to previous work:
>
>     We argue that rejecting wavelet-packets because Fourier and spatial approaches already exist is overly restrictive.
>     Consider smartphones, for example. If we applied the same rationale, we would have to reject them. After all, phones and cameras existed separately before.
>     Just as smartphones integrate cameras and phones into a new gadget, wavelet packets integrate space and frequency analysis into a new representation. We see value here!
>
> - Performance of the new approach:
>
>     Numerous papers exploring time or Fourier-based detection already exist (Dzanic et al.,2020; Frank et al., 2020; Durall et al., 2019; 2020; Giudice et al., 2021).
>     As we are the first to work in Wavelet-space, we had to lay the algorithmic groundwork. Making boundary wavelet transforms work in PyTorch is non-trivial and requires careful handling of the sparse operator matrices. Getting it right is a contribution to the broader community. We provide the source code, tests, and documentation for that purpose.
>
> - The wavelet-packet transform:
>
>     The repeated application of some or all four filters is the main difference between the fast wavelet transform and its packet variant. The choice is motivated by the desire to separate the high-frequency bands. Figures 1 and 3 yield some intuition regarding the effect of every filter. Without all four of our analyses would not span the entire spectrum.
>
> - Introduction to the perfect reconstruction and alias cancellation conditions:
>
>     Thank you for the suggestion. We have added a discussion in section 8.3 in the supplementary material.
>
> - Introduction to Daubechies Wavelets and Symlets:
>
>     Thank you for the suggestion. Supplementary section 8.4 now discusses both wavelets.
>
> - Comparison to other work:
>
>     We chose to work on the problem setup as presented by Frank et al. and Yu. et al. Both papers have been accepted at respected venues. Conducting experiments on entirely new datasets is not realistic for a rebuttal. We also decided to prioritize studying the robustness over the study of as many datasets as possible. In other words, we report mean and standard deviations over several seeds, even if this means we can study fewer datasets overall.
>
> - Model choice:
>
>     To illustrate the power of the wavelet packet approach, we choose the smallest possible networks. In the FFHQ/stylegan/stylegan2 case, working with linear networks on top of the packets is sufficient to obtain more than 98.5% accuracy. On CelebA and LSUN we required a CNN structure. Mainly to be competitive in comparison to Frank et. al 2020, and Yu et al 2019.
>
> - Resolution in Figure 1:
>
>     As stated in the last paragraph of page 3, we analyze 1024 by 1024 images in Figure 1. We have updated the caption to make this clearer.

---

> > ### Comment · Reviewer_feXB · 2021-11-29
> > **Reply to authors' rebuttal**
> >
> > The authors have clarified some of the points mentioned in the review. But I still think that just combining the Fourier and spatial approaches that already exist is not novel enough to accept this paper. The lack of testing on various deepfake datasets was a very important factor in regards to my previous decision. Although the authors claim to provide the source code for their work in work in Wavelet-space, the lack of novelty in the main approach is enough for me to stay on my previous decision. My final decision is "reject, not good enough".

---

> > > ### Author Response · Authors · 2021-11-30
> > > **Thank you for the honest verdict.**
> > >
> > > We understand the desire for more experiments and will add even more in a future version of this draft. However, we are not ready to accept the lack of novelty argument.
> > >
> > > Wavelets are fundamentally different from Fourier or Spatial approaches. Wavelets use both domains in a novel joint data representation and are therefore more than just simple concatenation.  This impression is probably also a miscommunication on our part. We will improve this in a future version of the paper.
> > >
> > > Using the current draft, we argue that figure one is novel since the observations therein have not been made and cannot be made by simply combining Fourier and Spatial approaches. Only wavelets reveal the locations of frequency differences. To make the argument against novelty convincing, we kindly ask for citations where something akin to Figure 1 has previously appeared?

---

### Official Review · Reviewer_qN3N · 2021-11-10

**Correctness:** 3
**Technical Novelty And Significance:** 3
**Empirical Novelty And Significance:** 2
**Recommendation:** 5
**Confidence:** 4

**Main Review:**

This paper proposes a very simple solution for deepfake detection based on wavelet representation.
Strength:
+ The wavelet has not been exploited in deepfake detection, which can provide multiscale representation from low-frequency representation to high-frequency representation.
+ From the experiments shown in this paper, the wavelet representation looks promising in deepfake detection, that many features at different stages can successful perform deepfake detection, and they achieve better results than image or DCT based representation.=
+ Overall, the paper is clearly written and easy to follow.

Weakness:
- The solution is trivial. It seems more study and designs can be exploited. Since wavelet provides multiscale representation, how can you leverage those features together for deepfake detection. However, this paper only simply learns a CNN to achieve classification. The idea of applying wavelet looks good and promising, but need a better solution for more favorable detection.
- The experiments are not solid. Even though authors test their methods on a set of GAN models, they don't test it on the synthetic images with attacks, such as cropping, JPEG compression, low-pass filtering etc. Besides, detecting GAN generated images is interesting, however, there are many existing deepfake detection benchmarks. The authors need to show results on the public benchmark and compare your method with more state-of-the-art models.
- References are not enough. The authors focus on discussing the image and frequency based representation. However, there are many other works in this area recently. Such as CVPR21, ICCV21, IJCAI21.
  * C. Wang et al., Representative Forgery Mining for Fake Face Detection, CVPR, 2021.
  * H. Zhao et al., Multi-Attentional Deepfake Detection, CVPR, 2021.
  * T.Zhao et al., Learning Self-Consistency for Deepfake Detection, ICCV, 2021.
  * Y. He et al., Beyond the Spectrum: Detecting Deepfakes via Re-Synthesis, IJCAI, 2021.

As a result, I think this work requires a lot of extensions for publication, and rate this submission as marginally below the acceptance threshold.




**Summary Of The Paper:**

This paper leverage wavelet-based representation in deepfake detection. By applying wavelet transformation, several features are obtained. In this work, the authors shows the wavelet features at different stages are more robust than previous widely used frequency or image based representation. The representation can be applied like image itself to perform deepfake detection, for example CNN and PRNU. This paper leverage synthetic images from different GAN models to conduct their experiments, including cross-domain detection from an unknown domain.

**Summary Of The Review:**

My main concerns are that the experiments are solid enough and the method is trivial. The authors only apply wavelet to extract multiscale features and test them separately.

---

> ### Author Response · Authors · 2021-11-19
> **We thank reviewer qN3N for the time to review our paper.**
>
>
> - Novelty of the method and quality of the writing:
>
>     We are happy to hear qN3N agrees with SHXs and feXB, who found our method novel and helpful. Furthermore, we were delighted to read that qN3N, like SHXs, and SoZC found our paper well written.
>
> - Related work:
>
>     Thank you for suggesting additional relevant sources. We have included all of them in the revised version of the paper.
>
> - Contribution and complexity of our approach:
>
>     We respectfully disagree with qN3N. We argue that the introduction of boundary wavelet packets into the deep-learning toolbox is significant. Getting the fast boundary wavelet transformation to work requires the careful creation of the necessary orthogonal sparse matrices. To assess the value of our work, we encourage qN3N to take a look at the source code, tests, and documentation.
>
> - Experimentation:
>
>     Thank you for pointing to weaknesses in our experimental section. We have run robustness tests and included the results in table 6. We find some resistance to center-crop and rotation perturbations.
>     We do compare to other work on the publicly available benchmark created by Yu et al., 2019 and Frank et al., 2020, and find competitive performance, even improving upon previous work on CelebA. We agree with qN3N adding additional results and benchmarks is always desirable. Yet, at the same time, we respectfully disagree with qN3N and argue that we do have enough data to show the utility of the packet approach.
>
> - On the relation of our work to the state of the art:
>
>     On the datasets, we have studied our method performed well in comparison to published work. We understand the reviewers desire to see more comparisons of additional datasets. However, we argue that our paper already contains enough data to show the potential of the proposed method.
>     According to the ICLR 2022 review guide ( https://iclr.cc/Conferences/2022/ReviewerGuide ), "state of the art" numbers are not the only way to create value for the community. We argue that adding boundary wavelet analysis to the ML-communities toolbox does help everyone since it opens up new analysis and detection pathways.

---

### Official Review · Reviewer_sHXs · 2021-11-10

**Correctness:** 4
**Technical Novelty And Significance:** 1
**Empirical Novelty And Significance:** 2
**Recommendation:** 3
**Confidence:** 5

**Main Review:**

The paper is well structured, mathematical details are well presented and experimental results demonstrate the effectiveness of the technique. However I have some concerns:

Section 4.2 and 4.3 presents results obtained on different image-sets. Please compare with more advanced SOTA techniques (as reported in related work)

Given the demonstration that first layers of CNNs encode something extremely similar to wavelets (Mallat, Stéphane. "Understanding deep convolutional networks." Philosophical Transactions of the Royal Society A: Mathematical, Physical and Engineering Sciences 374.2065 (2016): 20150203.) please provide explanation of the contribution of employing wavelet + CNN and not directly a CNN without the wavelet extraction phase.

Moreover the method does not take into account any kind of manipulation or transformation on images. How does compression affects the technique? Resizing? Rotation? Etc.

Many cited papers obtain better results in detection and classification. The authors compare for parameters. There are works that employs analytical solutions with extremely few parameters. They should be taken into account and improvements w.r.t. them should be emphasized

**Summary Of The Paper:**

This paper presents a wavelet-based approach for deepfake image detection. This approach is the first using wavelet for the task and could be of great contribution to the community mixing spatial and frequency information. Experimental results are presented for:

Binary classification (FFHQ vs. Stylegan 1/2)
CelebA+LSUN vs. GANs
Unknown generator test
Comparison with SOTA are carried out only vs. none of the best paper on the field

**Summary Of The Review:**

The paper is ok... but lacks of a contribution for community in my opinion and does not advance state of the art in any terms (computations speed, parameters, overall accuracy results, explaibaility).

---

> ### Author Response · Authors · 2021-11-19
> **We thank SHXs for reviewing our work**
>
> - Novelty:
>
>     Reviewer SHXs, in agreement with qN3N and feXB, found our paper to be the first application of wavelet packets for deepfake analysis and detection. Furthermore, we were happy to read that: " The paper is well structured, mathematical details are well presented and experimental results demonstrate the effectiveness of the technique. "
>
> We continue to argue that we create value by presenting a novel deepfake imagery analysis and detection approach.
>
> - Value to the community:
>
>     According to the ICLR 2022 review guide ( https://iclr.cc/Conferences/2022/ReviewerGuide ), "state of the art" numbers or not the only way to create value for the community. We argue that adding boundary wavelet analysis to the ML-communities toolbox does help everyone since it opens up new analysis and detection pathways. Numerous papers previously explored the use of the fast Fourier transform [ (Dzanic et al.,2020; Frank et al., 2020; Durall et al., 2019; 2020; Giudice et al., 2021)]. We are the first to use wavelet packets. Therefore, we had to lay the groundwork! Computing the boundary wavelet transform requires careful treatment of sparse matrix-operators. Making it work is significant. To make this point clearer, we have shared a link to an anonymous source code repository. Our automatic tests and documentation are easily accessible there.
>
> - Explainability:
>
>     We respectfully disagree with sHXs and argue that the wavelet package representation is interpretable. We showcase this feat in Figure 1 on page 3. Differences at the edges cause the high-frequency disparities. Localization of these differences was impossible with previously used purely Fourier-based methods.
>
> - Comparison to pure CNN:
>
>     As illustrated by table 4, using wavelet packets reduces the amount of training data we require. If the features of converged CNN resemble wavelets, why waste training data to learn them? We provide the means to save training time.
>
> - Experimentation:
>
>     Thank you for pointing to weaknesses in our experimental section. We have updated the paper. Table 6 now reports robustness test results. We find some resistance of our approach to center-crop and rotation perturbations.
>
> - Classification Performance:
>
>   In view of the quality of performance, we don't believe that a single representation or technique will be the approach to achieve the best performance in the end.   As mentioned, the introduced wavelet-based approach provides a competitive and interpretable addition, where we assume that an ensemble approach using ours and other approaches such as Fourier-based and directly on the pixels will be the way to go in future work.

---

### Author Response · Authors · 2021-11-12
**Anonymous source code repository**

Dear Reviewers, thank you for your feedback.

We noticed reproducibility and code quality did not appear in the reviews.
To make our source code, tests, and documentation more easily accessible, we have created an anonymous GitHub organization:

https://github.com/wavelet-detection-review

The Github organization hosts the code from the supplementary material.

We are working hard to address your feedback and will reply to each of you in detail as soon as we can.

---

### Public Comment · ~Moritz_Wolter1 · 2022-11-24
**Final Version**

We thank everyone for their feedback. The final version of this paper is now available at:
https://rdcu.be/cUIRt .

We share the source code via https://github.com/gan-police/frequency-forensics and https://github.com/v0lta/PyTorch-Wavelet-Toolbox .

---

### Decision · Program_Chairs · 2022-01-20

**Decision:**

Reject

**Comment:**

This paper received 5 quality reviewers, where 3 of them rated 5 and 2 rated 3. While this work has merits, many concerns are raised by various reviewers. The AC agrees with the reviewers that this paper is not ready for publication at its current form.